# Insights from DNA Barcodes-Based Phylogenetic Analysis of Medicinal Plants and Estimation of Their Conservation Status: A Case Study in the Tianshan Wild Forest, China

**DOI:** 10.3390/plants14010099

**Published:** 2025-01-01

**Authors:** Aerguli Jiamahate, Tohir A. Bozorov, Jiancheng Wang, Jianwei Zhang, Hongxiang Zhang, Xiyong Wang, Honglan Yang, Daoyuan Zhang

**Affiliations:** 1Key Laboratory of Ecological Safety and Sustainable Development in Arid Lands, Xinjiang Institute of Ecology and Geography, Chinese Academy of Sciences, Urumqi 830011, China; 18040921335@163.com (A.J.); tohirbozorov@yahoo.com (T.A.B.); www-1256@ms.xjb.ac.cn (J.W.); zjw18294830794@163.com (J.Z.); zhanghx561@ms.xjb.ac.cn (H.Z.); wangxy@ms.xjb.ac.cn (X.W.); 2University of Chinese Academy of Sciences, Beijing 100040, China; 3Laboratory of Molecular and Biochemical Genetics, Institute of Genetics and Plants Experimental Biology, Uzbek Academy of Sciences, Tashkent 111226, Uzbekistan; 4College of Agronomy, Xinjiang Agricultural University, Urumqi 830052, China

**Keywords:** analytic hierarchy process (AHP), DNA barcodes, medicinal plants, species identification, wild fruit forest

## Abstract

The Tianshan wild fruit forest region is a vital repository of plant biodiversity, particularly rich in the unique genetic resources of endemic medicinal plants in this ecological niche. However, human activities such as unregulated mining and excessive grazing have led to a significant reduction in the diversity of these medicinal plants. This study represents the first application of DNA barcoding to 101 medicinal plants found in the Tianshan wild fruit forests, using three genetic loci along with morphological identification methods. A phylogenetic analysis was performed to delineate species relationships. The results indicate that the internal transcribed spacer (ITS) region has been identified as the most reliable barcode for species identification across different families, while combining data from multiple gene segments can improve species detection. Moreover, the Analytical Hierarchy Process (AHP) was employed to assess and prioritize the 101 medicinal plants, highlighting 23 species as candidates for urgent conservation efforts in the region. The approaches and insights from this study provide a significant benchmark for DNA barcoding studies on medicinal plants with local significance and establish an evaluative framework for the conservation of biodiversity and the surveillance of genetic resources among medicinal plants in the Tianshan wild fruit forest area.

## 1. Introduction

Historically, medicinal plants were classified by botanists based on morphological characteristics, a method that is limited by the subtle nuances among plants with similar traits and the dwindling number of experienced taxonomists [1,2]. Currently, due to the subjectivity of morphological characteristics and differences in individual experience of taxonomists, as well as the incomplete taxonomic knowledge of global flora, these reasons may lead to inconsistencies in classification results, etc., thus seriously affecting work related to plant taxonomy [3]. DNA barcoding offers a solution to these challenges by providing a more reliable and efficient means of identification.

DNA barcoding, a revolutionary taxonomic tool, has emerged as a pivotal method for species identification since its inception in 2003. This technique utilizes standardized, short DNA sequences as markers, offering rapid, accurate, and automated identification capabilities that transcend the limitations of traditional morphological classification [4]. It is particularly valuable for non-specialists, as it is not constrained by the physiological conditions or morphological characteristics of the samples [5]. This method has been widely recognized and applied in the field of traditional Chinese medicine (TCM), which, with its rich history and regional diversity, is increasingly being integrated into mainstream healthcare practices due to its relative safety and efficacy [6,7].

The Internal Transcribed Spacer (ITS) is a component of ribosomal DNA, while *trnH-psbA* is a non-coding spacer found within chloroplast DNA. Both ITS and *trnH-psbA* serve as widely utilized molecular markers in plant DNA barcoding, particularly in flowering plants (angiosperms) [8]. These markers have demonstrated high discriminative power for species identification. When employed in tandem, they can enhance species identification accuracy, especially within certain species-rich genera [9]. However, while ITS exhibits relatively consistent performance across all plant types, its efficacy in gymnosperms is notably lower compared to angiosperms. Conversely, *trnH-psbA* displays less variability in some plant groups, suggesting that its application should be tailored to specific species [10]. Furthermore, ITS generally outperforms *trnH-psbA* in distinguishing between interspecific and intraspecific variation, thereby making it a more effective standalone marker [9]. Systematic investigations have shown that the plastid *matK* gene and the chloroplast *trnH-psbA* region, along with ITS, are suitable core barcode combinations for seed plants [11]. The Third World Congress on DNA Barcoding has further solidified *matK* and chloroplast gene segments, *rbcL*, as core sequences of plant DNA barcodes, guiding future research directions [12]. A study conducted by the Chinese Plant BOL Group on four genetic markers (ITS, *matK*, *trnH-psbA,* and *rbcL*) involving 1757 species from 141 genera has proposed that ITS/ITS2 serve as the primary barcodes for seed plants [13,14,15].

Given the slow evolution of mitochondrial genes in plants, the focus on plant barcodes has shifted to the chloroplast and nuclear genomes, which exhibit higher substitution rates [2]. The ideal single-locus barcodes for plants have yet to be identified, leading to the general consensus that multigene combination barcodes are the standard for plant barcodes [16,17,18,19]. It is essential to apply and compare multiple methods, such as Automatic Barcode Gap Discovery (ABGD) and Assemble Species by Automatic Partitioning (ASAP), which are, respectively, unsupervised and supervised learning operational classification methods [20,21]. The ABGD method is a clustering algorithm based on distance thresholds that identifies molecular operational taxonomic units (MOTUs) by setting various distance thresholds [22]. In contrast, ASAP employs an adaptive algorithm capable of dynamically adjusting its clustering criteria according to the phylogenetic distribution of samples [23]. This distinction in algorithms renders ASAP generally more flexible when addressing complex data sets. Regarding sensitivity to false matches, ABGD imposes higher requirements on input data and may be influenced by the quality and quantity of samples, whereas ASAP demonstrates greater robustness and can more effectively manage defects or inaccuracies in the data set [24]. Additionally, the ASAP method typically exhibits faster computation times, particularly when analyzing large data sets, making its speed advantage quite pronounced in scenarios where rapid results are essential [23,25,26]. Conversely, ABGD may necessitate longer computation times when processing larger or more complex data sets [23]. As two significant automatic clustering methods, ABGD and ASAP are pivotal in species classification and diversity assessment [26,27]. While they share certain similarities in goals, data processing methods, and computational efficiency, notable differences exist in computational efficiency, adaptability, methodological principles, and application contexts. ABGD is particularly suited for research demanding high accuracy, especially in the detailed classification of small-scale data sets, whereas ASAP is more appropriate for rapid assessments and extensive comparisons of species diversity, which is crucial in ecological monitoring.

Chinese herbal medicine, an integral branch of TCM, has been a cornerstone of healthcare in China for over 2000 years [28]. Xinjiang, known for its abundant biological resources and diverse climate, is home to a plethora of medicinal herbs, with 113 species of medicinal plants identified [29]. As the global trend toward valuing TCM and natural medicine gains momentum, the international community is increasingly recognizing the importance of medicines and health foods derived from natural herbal resources. This has led to a growing emphasis on the identification and conservation of herbal plants, which are crucial for biodiversity conservation and the safe utilization of these resources [30].

The Tianshan Wild Fruit Forest region, located in the Central Asian desert hinterland, is one of the few natural wild fruit forests in the world and a botanical hotspot for plant diversity [31]. Despite the presence of protected areas, the flora in the highlands faces constant threats from deforestation and overgrazing. Conducting a comprehensive and systematic assessment of the herbaceous plant resources in the area using rigorous scientific methods is of utmost importance. In terms of species protection decision-making, a comprehensive evaluation system for plant protection and utilization is built based on the AHP analytic hierarchy process to quantify and compare different factors of species from multiple dimensions such as biological characteristics, ornamental characteristics, and development potential [32]. Based on the quantified comprehensive evaluation of plants The evaluation values are graded to make accurate and effective development and utilization decisions for plants of different grades [33].

This work presents the findings from a study on local medicinal plants in the wild fruit forests of the Tianshan Mountains. Employing molecular biology methods, the study aimed to accurately identify the species and assess the feasibility of using ITS, *rbcL*, *matK,* and their combinations. Given the irrational exploitation of medicinal plants in the region, their endangered status, and the unsustainable exploitation of their natural habitats, it is essential to evaluate the current medicinal herbs using an effective multilevel evaluation system. This approach aims to provide reliable data for local institutions to implement effective conservation measures.

## 2. Results

### 2.1. Morphological Observation and Identification

This is the first systematic study in the wild apple region of the Tianshan Mountains conducting multi-loci marker analysis to delineate the boundaries of native wild medicinal plants (Figure 1A). A total of 101 plant species were identified (Appendix A), representing 83 genera and 36 families (Figure 1B). Of these, 29.36% (32 species) are endemic. The growth forms of these species included 81 herbaceous species (80.20%), consisting of 72 perennial herbs and 9 annual herbs, 12 shrub species (11.88%), 2 vine species (1.98%), 6 tree species (5.94%). This is consistent with previous studies, which show that most medicinal plants are perennial [34]. To assess the threat status of terrestrial plants in China and provide a scientific basis for implementing conservation measures aimed at mitigating the risk of plant species reduction and extinction, as well as protecting and managing China’s rich plant diversity, the Chinese Academy of Sciences and relevant research institutions released the “Red List of Higher Plant Biodiversity” and the “Red List of Vertebrate Biodiversity” in 2013 and 2015 [35,36]. This marked a significant step in China’s efforts to protect biodiversity. According to the standards of the International Union for Conservation of Nature (IUCN), plants are classified into various threat levels, including Critically Endangered (CR), Endangered (EN), and Vulnerable (VU) [35]. As of 2020, the inventory includes 39,330 species of terrestrial plants, encompassing subspecies units [37]. The assessment of medicinal plants for the IUCN Red List revealed that 12 species were not listed, while 89 species were included in the Red List. Among these, one species is categorized as endangered, five as near-threatened, two as data deficient, and 81 as least concern (Figure 1C). Further analysis demonstrated that among the 101 medicinal plants, the top 6 families were Asteraceae (15 species), Lamiaceae (11), Rosaceae (9), Fabaceae (8), Apiaceae (6), and Ranunculaceae (6) (Figure 1C). These results are consistent with those of Yu et al. who also identified the top 10 families with the highest number of medicinal plants [38]. Additionally, the study identified 39 habitat types, which were diverse and included stony slopes, subalpine grasslands, coniferous forest scrub meadows, river sands, spruce understory, and others (Appendix A).

### 2.2. PCR and Sequence Analysis

Total DNA was extracted from 109 samples representing 101 medicinal plant species collected from various wild fruit forests in the Tianshan mountains. These samples were subjected to direct sequencing. A total of 327 sequences were obtained, with 100% PCR amplification efficiency for each of the ITS, *matK,* and *rbcL* (Table 1). The characteristics of the sequences for the three regions are summarized in Table 1. The ITS sequence length ranged from 278 to 860 bp, with a GC content of 48.6–66.6% and nucleotide diversities (π) of 0.3. There were 116 variable sites identified, which were potentially polymorphic. Similarly, the *matK* sequences ranged from 662 to 926 bp in length, with a GC content of 28.9–38.2% and a Pi of 0.38, including 408 variable sites. The *rbcL* sequences ranged from 567 to 722 bp, with a GC content of 41.80–45.70% and Pi of 0.1, containing 397 polymorphic sites. Consistent with previous studies indicating that DNA stability increases with higher GC content in the ITS region [39]. The sequence characteristic observed in this study aligns with these findings. The relative GC content was as follows: ITS > *rbcL* > *matK*. Based on the number of variable sites and parsimony-informative sites, the variability decreased in the order: *matK* > *rbcL* > ITS. When comparing π values, *matK* showed the highest value, followed by ITS and *rbcL*. Sequences obtained from *rbcL* exhibited the highest percentage of conserved sites (25.8%), followed by ITS (9.7%) and *matK* (3.1%).

### 2.3. Genetic Distance Analyses

Pairwise genetic distance analysis was performed within and among species using all three DNA regions (ITS, *matK*, and *rbcL*) by employing the Kimura 2-parameter (K2P) model with 1000 bootstrap replications [4,40]. Among the three individual barcode regions and four combinations, *matK* + *rbcL* exhibited the highest mean intraspecific genetic distance (0.599), followed by ITS + *rbcL* (0.462), *matK* (0.168), ITS + *matK* (0.124), ITS (0.096), ITS + *matK* + *rbcL* (0.078), and *rbcL* (0.008). For inter-species distances, the ranking in descending order was as follows: *matK + rbcL* (0.678), ITS (0.676), *matK* (0.626), ITS + *rbcL* (0.621), ITS + *matK* (0.616), ITS + *matK* + *rbcL* (0.568), and *rbcL* (0.123) (Table 2). An analysis of barcoding gaps revealed apparent gaps in the ITS, *matK*, and ITS + *matK* regions, although slight overlaps in intra- and inter-specific genetic distances were observed in these markers. For other single markers and certain combinations of candidate loci, no distinct barcoding gaps were found, and the overlapping distributions of intra- and inter-specific variation were evident (Table 2 and Appendix A and Figure 2).

### 2.4. Species Discrimination Using the BLAST Method

Notable similarities in taxonomic patterns were observed across the three regions in terms of species identification rates based on BLAST analysis (Figure 3). Several species-rich families, such as Asteraceae, Lamiaceae, Rosaceae, Fabaceae, and Apiaceae, exhibited low species-level identification rates in each region. The pairwise identification (PI, %) at the species level ranged from 97.2% to 100% for ITS, 98.6% to 100% for *matK*, 99% to 100% for *rbcL*, 99% to 100% for ITS + *matK*, 98.3% to 100% for ITS + *rbcL*, 82.8% to 100% for *matK* + *rbcL*, and 99% to 100% for ITS + *matK* + *rbcL* (Figure 3A). The identification rate results demonstrated that the resolution at the family level (93.58~98.10%) was superior to that at the genus level (90.74~94.55%) and the species level (61.39~74.65%) (Figure 3B–D; Appendix A). Overall, the combination of multiple loci significantly enhanced the accuracy of species identification.

### 2.5. Distance-Based Species Delimitation

To group specimens into molecular operational taxonomic units (MOTUs) and predict species boundaries, we employed two species delimitation methods: the ABGD and ASAP algorithms. ABGD relies solely on pairwise genetic distances and does not necessitate the construction of a phylogenetic tree. In contrast, ASAP employs a hierarchical clustering algorithm. While ASAP also depends on pairwise genetic distances, its structure and implementation are more complex than those of ABGD. For a diverse range of species across large areas, ABGD may be more efficient, whereas ASAP is better suited for finer classification. ABGD prioritizes differences in genetic distance, while ASAP organizes groups based on sequence similarity. Together, these methods complement each other, enhancing the accuracy of species identification. Given that the species in this study are part of a large-scale multi-species context, both methods can generate candidate lists for species partitioning. The analysis revealed varying numbers of MOTUs, specifically: 96/84 MOTUs for ITS, 65/103 MOTUs for *matK*, 90/97 MOTUs for *rbcL*, 103/93 MOTUs for the combined ITS + *matK*, 100/89 MOTUs for the combined ITS + *rbcL*, 42/97 MOTUs for the combined *matK* + *rbcL*, and 90/97 MOTUs for the combined ITS + *matK* + *rbcL*, as delimited by the ABGD and ASAP methods, respectively (Figure 4, Appendix A). Furthermore, in terms of species identification rates, the ABGD method performed as follows: ITS + *matK* (94.50%), ITS + *rbcL* (91.74%), ITS (88.07%), *rbcL* (82.57%), ITS + *matK* + *rbcL* (82.57%), *matK* (59.63%), *matK* + *rbcL* (38.53%), while the ASAP method is *matK* (94.50%), *rbcL* (88.99%), *matK* + *rbcL* (88.99%), ITS + *matK* + *rbcL* (88.99%), ITS + *matK* (85.32%), ITS + *rbcL* (81.65%) and ITS (77.06%) (Appendix A).

### 2.6. Phylogenetic Tree-Based Species Identification

A phylogenetic tree’s construction and the evaluation of monophyly rates based on molecular data represent the simplest approach to gauge a species’ DNA barcode’s ability to discriminate [40]. To enhance the reliability and accuracy of species identification, we constructed trees using maximum likelihood (ML) for each candidate barcode, along with the highest-hit sequences obtained from BLASTn in GenBank. The inferred distance-based species phylogenetic tree is illustrated in Figure 5 and Appendix A. The species trees from various sites were well-resolved and demonstrated strong support, with the majority of branches exhibiting maximum support values. This finding aligns with traditional taxonomic classification methods. In particular, clustering was tighter for single barcodes than for combinations of barcodes. Additionally, several Lamiaceae species were observed to be sister taxa to species from other families in all trees (Figure 5 and Appendix A). To further investigate species monophyly at each sampled locus and at the combined loci, we tested the conspecific clusters of the species. The results revealed that most species were conspecific clusters at all or almost all loci, 70.30% of species were conspecific clusters at ITS loci, which were 62.38% at *matK* locus, and 59.41% at the *rbcL* locus. Notably, the monophyly rate for combined loci was lower than that for individual loci, with the following proportions: 32.67% (ITS + *matK*), 38.61% (ITS + *rbcL*), 39.60% (*matK* + *rbcL*), and 26.73% (ITS + *matK* + *rbcL*) (Appendix A). However, the clustering of the seven trees highlighted significant taxonomic inconsistencies. All single-locus phylogenetic trees were more effective at forming conspecific clustered branches, demonstrating strong species discrimination. A comparison of phylogenetic trees based on various gene combinations revealed confusion and inconsistency in species identification across this diverse set of species. These findings further confirm that an in-depth exploration of the phylogenetic signals from different genomic regions across multiple species can elucidate the complex evolutionary history and relationships among species. Therefore, it is essential to establish a regional database of medicinal plant DNA information that covers a broader range of species.

### 2.7. Evaluation of Medicinal Herbs

Scores of 101 medicinal plants were calculated using the AHP method. Based on the composite evaluation value from multiple indicators, the species were ranked into three classes. Specifically, 10 plants were classified as Class 3, 68 were classified as Class 2, and 23 plants were in Class 1 (Appendix A). Among the *Prunus armeniaca*, *Glycyrrhiza uralensis*, *Rhodiola quadrifida*, and *Pseudolysimachion alatavicum* were identified as near-threatened species. The AHP scores ranged from 0.28 to 0.69, with 10% of the plants falling into level 3 (0.28~0.39), 67% into level 2 (0.40~0.60), and 23% into level 1 (0.6~1.0). The 23 species classified as level 1 were distributed across the following: Apiaceae (1), Asteraceae (4), Betulaceae (1), Boraginaceae (1), Crassulaceae (1), Fabaceae (3), Lamiaceae (3), Liliaceae (1), Papaveraceae (1), Plantaginaceae (1), Polygonaceae (1), Primulaceae (1), Ranunculaceae (2), and Rosaceae (2), including four near-threatened species within Class 1. Regarding endemism, 19 species were found to be endemic to Xinjiang (Table 3, Figure 6). In the AHP method assessment, species were evaluated from ecological, economic, and genetic perspectives. Species of particular ecological importance included *Gagea serotina*, *Primula algida*, *Lathyrus tuberosus*, *Inula racemosa*, *Ligularia heterophylla*, *Oxytropis ochroleuca*, *Roemeria refracta*, *Thymus marschallianus*, *Thymus proximus, Rosa laxa*, and *Betula tianschanica*. Medicinal plants, including *Rheum wittrockii*, *Aquilegia atrovinosa*, *Cynoglossum officinale*, *Glycyrrhiza uralensis*, and *Rhodiola quadrifida*, possess distinct medicinal properties and active compounds, contributing significantly to the economic value of pharmaceutical and healthcare products. From a genetic standpoint, *Leonurus turkestanicus*, *Doronicum altaicum*, *Arctium tomentosum*, *Conium maculatum*, *Pseudolysimachion alatavicum*, *Prunus armeniaca* and *Aconitum nemorum* were identified as being of particular interest including population genetic structure, genetic diversity of germplasm resources, genetic differentiation and adaptability, etc. [41,42,43].

## 3. Discussion

Xinjiang is one of the richest regions in the world in terms of medicinal plant resources in China. Research indicates that over 1000 species of medicinal plants have been documented in this region, of which approximately 450 are the most commonly utilized. The use of these plants has a long-standing history and is often intricately linked to local culture, customs, and daily life. Reports suggest that the region possesses a rich medicinal tradition that has incorporated these plants for thousands of years [44]. In-depth research on medicinal plants in Xinjiang has gradually unveiled that many unique species in the region possess medicinal properties, with their active ingredients demonstrating significant effects in immune regulation, antioxidation, antitumor activity, and other areas [45]. The Tianshan wild fruit forest area is a crucial distribution zone for medicinal plants in Xinjiang and has yielded a variety of medicinal plant resources with distinct regional characteristics. This area is characterized by a high concentration of species within a limited number of families, encompassing 60 families, 237 genera, and a total of 435 species. Among these, there are nine species classified as national key protected plants, three species recognized as rare and endangered in China, and nine species designated as key protected plants within the autonomous region. Furthermore, the area supports 131 species of medicinal plants. The distribution of these species underscores the uniqueness of the wild fruit forest plant resources and highlights the transitional nature of the local flora [29].

Research on the population genetics and endophyte biological control functions of certain medicinal plants in the wild fruit forest area of Tianshan Mountains in Xinjiang shows the uniqueness of medicinal plants in this area [46,47]. Therefore, systematic research on medicinal plants in this area is extremely necessary and important. However, over the past 20 years, there has been a notable lack of updated research on medicinal plants in this region, particularly regarding systematic studies at the molecular biology level. Consequently, there is a pressing need for more in-depth and systematic research on molecular biological systems in the Tianshan Mountains. Such research is essential for understanding the genetic diversity, adaptability, and potential medicinal value of these plants. Compared with previous studies, this study examined 59 medicinal plants in addition to those included in the regional medicinal plant list, thereby expanding the regional medicinal plant resource base [29]. This expansion is of significant importance for the protection of biodiversity and the sustainable utilization of wild fruit forest areas in the Tianshan Mountains of Xinjiang. The newly added species not only enhance the regional medicinal plant database but also offer additional genetic resources and research materials for future studies in medicinal plant research.

Despite Xinjiang’s rich resources of medicinal plants, the status of their protection is concerning. Currently, the habitats of many medicinal plants are under threat, primarily due to over-collection and habitat loss [48]. Consequently, there is an urgent need for comprehensive research on the distribution and protection status of medicinal plants in Xinjiang to develop effective conservation measures. Plant classification is a vital branch of biology that aids scientists in understanding the diversity, evolutionary relationships, and ecological functions of plants [3]. Implementing appropriate classification systems ensures global consistency in the naming and categorization of plants, which is essential for the scientific research, development, and conservation of medicinal plants [49]. The classification of medicinal plants often relies on morphological characteristics. However, this approach may encounter limitations when addressing certain similar species. The advent of genomics technology has provided new insights into the study of genetic diversity among these plants. In this study, methods such as ASAP, ABGD, BLAST, and phylogenetic tree analysis were employed to assess the species discrimination capabilities of multi-locus DNA barcodes, including ITS, *matK*, *rbcL*, and various combinations of these regions in medicinal plants. The results indicated that the ITS region comprehensively exhibits a superior species discrimination rate, suggesting it is a potentially ideal barcode region for identifying medicinal plants in the Tianshan wild fruit forest area. This result validates the findings of numerous studies on medicinal plant identification, supports the use of ITS/ITS2 sequences as essential DNA barcodes for seed plants, and further reinforces the primacy of ITS as a DNA barcode specifically for medicinal plants [13,15,16,50,51]. Notably, in our analysis of the discriminative power of medicinal plant barcodes in this region, we found that the combined use of *matK* and *rbcL* exhibited relatively low discriminatory power compared to other multilocus combinations. This finding is somewhat inconsistent with the conclusions reached by the CBOL Plant Working Group on the *matK* + *rbcL* plant barcode. In addition, the discriminative power of the ITS + *matK* + *rbcL* combination was found to be inferior to that of individual barcodes [52]. This difference may be due to the wide range of species involved in this study as well as species uniqueness. When identifying species, the lack of species reference sequences in relevant public databases resulted in incomplete data or insufficient sample coverage for the ITS + *matK* + *rbcL* combination. Combinations may not cover variation in all species, resulting in reduced discriminatory power.

After confirming the correct species identification through both morphological and molecular biological methods, we conducted a preliminary discussion on the priority conservation assessment system for key wild plant genetic resources of medicinal plants in the region. This study focused on medicinal plants within the wild fruit forest area of the Tianshan Mountains as a case study, employing the Analytical Hierarchy Process (AHP) to evaluate 101 medicinal plants from the perspective of comprehensive conservation and utility. The evaluation system comprised two standard layers and thirteen index factors. The plants were categorized into three levels based on their comprehensive scores, with the first-level category consisting of 23 plants identified as wild plant genetic resources prioritized for conservation in the region. The Tianshan wild fruit forest serves as a globally significant gene bank for economic fruit tree resources. In recent years, this forest has undergone extensive degradation due to the impacts of climate change and human activities. To safeguard this vital natural genetic library, it is essential to establish an evaluation system that scientifically assesses the current status of wild plant genetic resources, monitors their changing trends in a timely manner, and reflects both the reserves and the utilization value of these resources. The AHP can be utilized as a tool to evaluate and prioritize various conservation and management strategies, thereby ensuring the long-term sustainability of these medicinal plant resources [53,54]. Based on the AHP evaluation, we advocate for the adoption of differentiated management strategies tailored to various levels of plant resources to ensure their effective protection and rational utilization. Specifically, for first-level protected plants, we propose conducting in-depth research on genetic diversity and establishing a molecular identification system. For second-level plants, we recommend enhancing periodic monitoring of population dynamics to promptly assess changes in their growth status and distribution. Regarding third-level plant resources, we encourage rational development and utilization, ensuring ecological safety as a prerequisite, and advocate for the sustainable development and utilization of medicinal plant resources, guided by market principles, to achieve a mutually beneficial outcome of economic gains and ecological protection. Additionally, for medicinal plant resources unique to Xinjiang, such as *Glycyrrhiza uralensis* and *Pseudolysimachion alatavicum*, along with other rare species, we strongly recommend implementing protective fencing, strictly regulating mining activities, and fostering scientific and technological innovation to enhance resource utilization efficiency. On this basis, we should actively explore an industrial model that balances protection and development, with the aim of achieving the sustainable protection and utilization of medicinal plant resources in the wild fruit forest area of the Tianshan Mountains. This study advocates for the development of tailored management strategies for plant resources across varying levels of protection to ensure their conservation and sustainable use. In all, for the first-class medicinal plants evaluated by APH in this article, it is proposed that in-depth genetic research and molecular identification systems be implemented to enhance protection and guide scientific utilization. The second-level medicinal species necessitate improved monitoring of population dynamics to effectively track changes in growth and distribution. Third-level medicinal resources should be developed and utilized rationally, and particularly those medicinal plants that are currently in widespread use should be managed sustainably in accordance with market principles to achieve both economic and ecological benefits.

Although techniques such as molecular biology have been extensively utilized in the classification and research of medicinal plants, the integration of emerging technologies with traditional morphological taxonomy remains a significant challenge. In addition, the identification of many medicinal plants has been well documented, while the systematic application of molecular biology and deep learning techniques in conjunction with morphological taxonomy is lacking, resulting in instances of misclassification or the oversight of certain plants. Therefore, by emphasizing species identification that combines plant morphological taxonomy with molecular biology methods, the subsequent identification of medicinal plants in Xinjiang can be made more efficient and widespread. What is more, in the application of DNA barcoding, there are cases where the number of samples collected in certain biological regions is insufficient, resulting in the inability to accurately identify plant species in these areas. Therefore, it is necessary to strengthen the construction of regional plant DNA barcoding databases covering more species and genetic markers.

## 4. Materials and Methods

### 4.1. Study Area

In this study, we randomly collected 109 medicinal plants along Tianshan Wild Forest Mountain in Xinjiang, China, which has a wide distribution in Xinjiang. In total, 109 vouchers representing 101 native wild plant species were collected, denoting 83 genera and 36 families over the course of 4 years in this field of experiment. Herbarium vouchers for all analyzed species, with the exception of threatened species, are deposited in the herbarium of Xinjiang Institute of Ecology and Geography, Chinese Academy of Sciences. The identification of plant specimens was performed by taxonomists of our institute, and detailed information on the voucher specimens is provided in Appendix A. A map of the spatial distribution of the collection sites of this study was produced using ArcGIS 10.3 software.

### 4.2. DNA Extraction and Amplification

Total DNA was extracted from 0.2 g of silica gel-dried plant leaf material using the DNeasy Plant Mini Kit (Qiagen, Hilden, Germany) according to the manufacturer’s instructions. DNA concentration and purity were assessed by measuring the absorbance ratio at 260 nm and 280 nm, ensuring that the ratio ranged from 1.7 to 1.9. DNA integrity was evaluated by agarose gel electrophoresis (agarose gel concentration: 1%; voltage: 120 V; electrophoresis time: 20 min). The nrITS and two plastid DNA (ptDNA) markers (*matK* and *rbcL*) were amplified and sequenced using the primers listed in Table 1. The PCR was conducted in a 30-μL mixture including 15 μL of 2× Taq PCR Master Mix (Applied Biosystem, Carlsbad, CA, USA), primer pairs 2 μL (5 pmol/μL) each (Table 4), 1.0 μL of DNA template (about 20 ng), and 10 μL of ddH_2_O (Table 4). The thermal cycling conditions were as follows: 95 °C for 5 min and 35 cycles of 95 °C for 30 s, optimal annealing temperature (55 °C for ITS, 50 °C for *matK*, 60 °C for *rbcL*) for 30 s, and 72 °C for 45 s, followed by 72 °C for 10 min. ABI 3730xl DNA analyzer (Applied Biosystems) was chosen to resolve double-strand sequence products at Huayu Gene (Wuhan, China) using PCR primers as sequencing primers.

### 4.3. Sequence Editing and Phylogenetic Analyses

To confirm overall sequence quality, Chromas v 2.6.6 (http://technelysium.com.au/wp/chromas, accessed date: 31 October 2024) was used to view trimmed and edited raw nucleotide sequences. The paired-end reads were processed and merged using Geneious (Geneious Prime v11.0.18, Biomatters). Given the wide range of species included in the analysis, the MAFFT algorithm in Geneious Prime, which is suitable for processing large data sets, was used to facilitate sequence alignment using various algorithms. In addition, manual sequence pruning was performed on regions that presented alignment challenges, thereby improving the accuracy of the comparison. To confirm the taxonomic identification of the derived sequences, a BLAST search against the NCBI Nucleotide Collection (nr/nt) database was performed to identify these representative sequences [55]. The original sequences were queried in turn to determine whether the closest results were homologous, with an E-value threshold < 1 × 10^−5^ [55]. Species identification of an individual is considered successful when the queried sequence has the highest matching hit with conspecific individuals’ binomial names [56]. To differentiate the species based on the genetic distance cut-offs, all datasets aligned by MAFFT were used to generate the genetic distances using MEGA based on p-distance, and the mean intraspecific and interspecific results were plotted with Prism 9.0 [57,58]. In terms of genetic distance judgments for species delimitation, identification is deemed successful when the maximum intra-species distance is less than the minimum inter-species distance for each species [12,59]. To further evaluate species delimitation hypotheses, Automatic Barcode Gap Discovery (ABGD) and Assemble Species by Automatic Partitioning (ASAP), both distance-based delimitation methods, were employed to conduct species delimitation on the alignment of single barcode marker sequences and a concatenated data set, using the web server with default settings. To investigate the phylogenetic relationship between species, the maximum-likelihood (ML) phylogenetic tree was constructed using IQ-TREE2 v.2.1.2 with the best-fit substitution model selected using the ModelFinder algorithm and support values assessed using the UFBoot approach with 1000 replicates [60,61]. In the phylogenetic trees, the fact that all individuals of conspecifics formed a conspecific clade was regarded as a successful distinction between species with UFBoot support values > 95% support corresponding roughly to a 95% probability that a clade is true [60,62].

### 4.4. Comprehensive Evaluation and Analysis of Medicinal Herbs by AHP

To scientifically estimate the current status of wild medicinal plants in the region, the Analytical Hierarchy Process (AHP) was used to rank the investigated species. A systematic evaluation model with two to five first-level indicators (Index layer) and thirteen second-level indicators (criterion layer) was constructed. According to the 1–9 scale method of binary relative comparison (Table 5), values were assigned to the indicators at each level of the standard and constraint layers, and weights were calculated based on the resulting judgment matrix (Table 5). These criteria and scores were determined based on numerous social surveys and the combined opinions of experts and authors in the relevant fields. This evaluation system scientifically and effectively assessed the actual status of the medicinal plants collected in this study, providing reference and suggestions for the priority conservation assessment of these important wild medicinal plants. The evaluation criteria were categorized in Appendix A.

## 5. Conclusions

This paper studied the list of medicinal plants found in the Tianshan wild fruit forests, identifying 101 medicinal plants with therapeutic value. This research further enriches the diversity of medicinal plants in the region. At the same time, the study reveals that ITS can be used as a core barcode for medicinal plants in this region, while multi-locus combinations have lower species discrimination. In addition, for species identification in this region, we recommend using multiple methods that combine morphological taxonomy and molecular biology for scientific verification, thereby improving the efficiency of species identification. Based on this foundation, we developed a scientific assessment system for the genetic resources of 101 medicinal plant species. We also proposed tailored protection strategies for plants categorized at various assessment levels, with particular emphasis on the 23 species identified at the highest level in this article. These findings may help explore the genetic diversity, adaptability, and potential medicinal value of medicinal plants in the Tianshan wild fruit forest area, and also provide strong support for advancing species taxonomy and biodiversity conservation.

## Figures and Tables

**Figure 1 plants-14-00099-f001:**
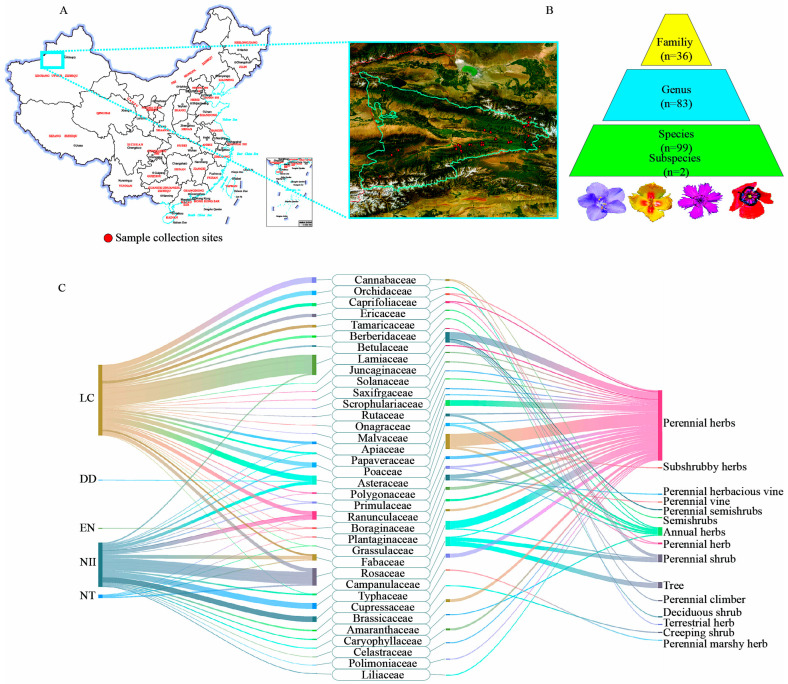
Composition and statistics of medicinal plants in Tianshan wild forest. (**A**) Sample sites for the medicinal species included in the present study. The map was created using ArcGIS 10.3 software. Red dots indicate the collection sites. The Map Content Approval Number: GS(2019)1682. (**B**) Medicinal plant community composition. (**C**) Assessment of medicinal plants for the IUCN Red List, family composition of medicinal species, and habitat types of these species across different family vouchers. EN represents endangered; DD represents data deficient; NT represents near threatened; LC represents least concern; NII represents not included in the IUCN Red List.

**Figure 2 plants-14-00099-f002:**
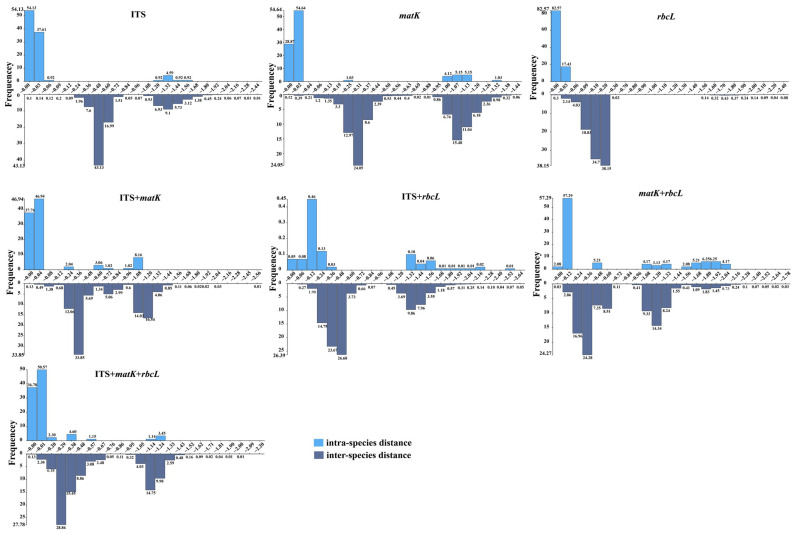
Barcoding gaps of each single candidate barcode and their combination markers based on intra- and inter-specific distances for species. Among them, there are 215 sequences of ITS sites (109 sequences in this study, 106 NCBI sequences downloaded), 216 sequences of *matK* sites (109 sequences in this study, 107 NCBI sequences downloaded), and 215 sequences of *rbcL* sites. (109 sequences in this study, 106 NCBI sequences downloaded), a total of 215 sequences of ITS + *matK* sites (109 sequences in this study, NCBI sequence download 106), ITS + *rbcL* site 216 sequences (109 sequences in this study, NCBI sequence download 107), *matK* + *rbcL* site 215 sequences (109 sequences in this study, NCBI sequence download 106 bar), a total of 214 sequences of ITS + *matK* + *rbcL* sites (109 sequences in this study, 105 NCBI sequences downloaded). Please see the Appendix A for detailed download sequences.

**Figure 3 plants-14-00099-f003:**
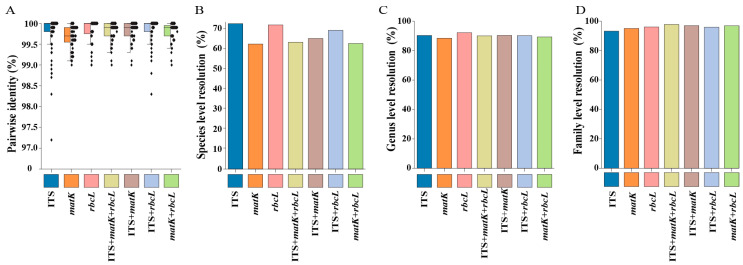
Species discrimination based on the similarity-based method (BLAST). (**A**) Pairwise identification success for each locus and multi-loci markers in this study. (**B**–**D**) Identification success rates at the species level, genus level, and family level using BLAST.

**Figure 4 plants-14-00099-f004:**
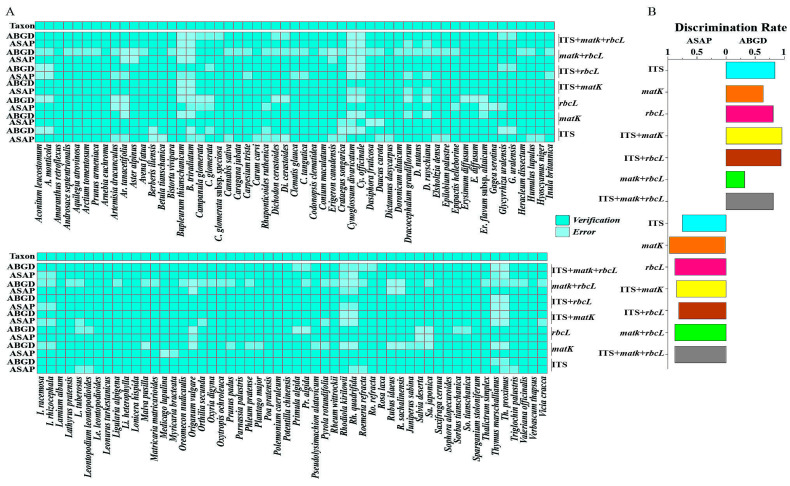
Species delimitation determined by ABGD and ASAP methods. (**A**) Taxon: morphological identification. (**B**) Species discrimination rates based on ABGD and ASAP. The delimited MOTUs by ABGD and ASAP analyses are shown.

**Figure 5 plants-14-00099-f005:**
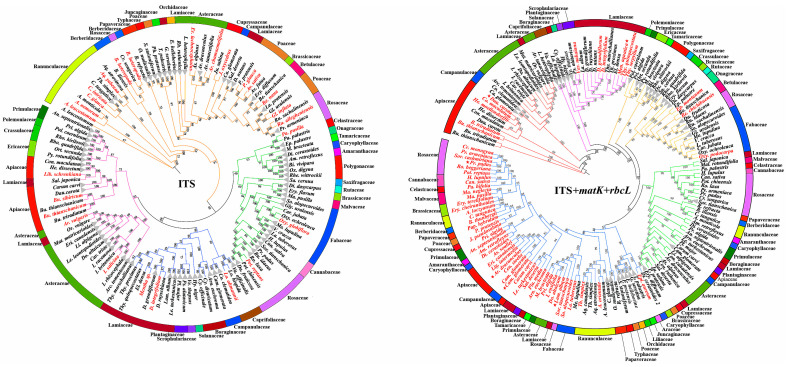
The Maximum likelihood (ML) tree of the medicinal species. Trees were constructed based on the *ITS* regions for 109 samples and other species sequences were downloaded via BLASTn from NCBI, as well as the following combinations: *ITS* + *matK* + *rbcL*. Bootstrap values are shown on the branches, ranging from 50% to 100%. Red fonts represent the sequences downloaded from NCBI via BLASTn and stars represent the subspecies sequence downloaded from NCBI: *Daucus carota* subsp. *maximus*, *Mentha aquatica* var. *citrate*, *Lamium album* subsp. *album*, *Campanula glomerata* var. *dahurica*, *Artemisia dracunculus* var. *glauca*, *Prunus padus* subsp. *borealis*. Gray triangles show the conspecific clustered branches including our samples and the downloaded sequences from NCBI via BLASTn which belong to the same species and only species names were labeled after the gray triangles. Detailed monophyletic species and the whole clustering information of all sequences are shown in Appendix A. The different colors of the branches in the trees represent the different clusters.

**Figure 6 plants-14-00099-f006:**
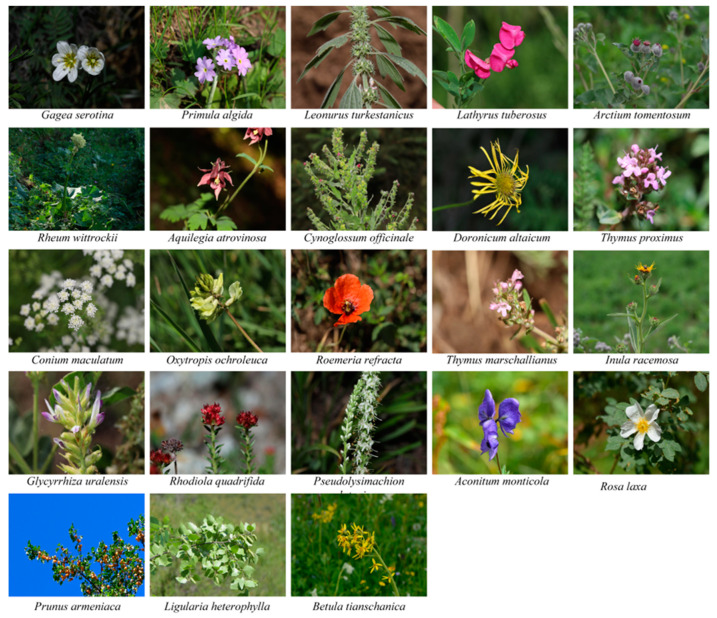
The morphological characteristics of species ranked in Class 1 by the hierarchical analysis (AHP).

**Table 1 plants-14-00099-t001:** PCR amplification and sequencing success rate, and sequence characteristics of each candidate barcode.

Items	ITS	*matK*	*rbcL*
Number of samples (individuals)	109	109	109
Success rates for PCR amplification (%)	100	100	100
Success rates for sequencing (%)	100	100	100
Length range (bp)	278–860 (678)	662–926 (799)	567–722 (591)
GC content (%)	48.6–66.6 (56.4)	28.9–38.2 (33.86)	41.8–45.7 (43.81)
Polymorphic sites	116 (10.17%)	408 (35.17%)	397 (51.36%)
Nucleotide diversity (π)	0.302	0.38	0.10
Non-parsimonious variable sites	112 (9.82%)	392 (33.79%)	191 (24.71%)
Aligned length (bp)	1141	1160	773
Conserved sites	70 (9.7%)	24 (3.1%)	151 (25.8%)

**Table 2 plants-14-00099-t002:** Summary of the pairwise intra-specific and inter-specific distances in the barcodes.

Barcode Locus	Intra-Species Distance			Inter-Species Distance		
	Minimum	Maximum	Mean	Minimum	Maximum	Mean
ITS	0.000	1.479	0.096	0.000	2.434	0.676
*matK*	0.000	1.298	0.168	0.000	1.434	0.626
*rbcL*	0.000	0.009	0.0008	0.000	2.352	0.123
ITS + *matK*	0.000	1.078	0.124	0.000	2.557	0.616
ITS + *rbcL*	0.000	2.432	0.462	0.000	2.632	0.621
*matK* + *rbcL*	0.000	2.034	0.599	0.000	2.771	0.678
ITS + *matK* + *rbcL*	0.000	1.195	0.078	0.003	2.193	0.568

**Table 3 plants-14-00099-t003:** Top 23 species ranked by a hierarchical analysis of hierarchy (AHP) in Class 1.

Species	Red List	Family	AHP Value	Endemic to Local (Y/N)	Ecological Type
*Gagea serotina*	-	Liliaceae	0.60	N	Perennial Herb
*Primula algida*	LC	Primulaceae	0.60	Y	Perennial Herb
*Leonurus turkestanicus*	LC	Lamiaceae	0.60	Y	Perennial Herb
*Lathyrus tuberosus*	LC	Fabaceae	0.60	Y	Perennial Herb
*Inula racemosa*	LC	Asteraceae	0.60	Y	Perennial Herb
*Ligularia heterophylla*	LC	Asteraceae	0.60	Y	Perennial Herb
*Rheum wittrockii*	LC	Polygonaceae	0.60	Y	Perennial Herb
*Aquilegia atrovinosa*	LC	Ranunculaceae	0.60	Y	Perennial Herb
*Cynoglossum officinale*	-	Boraginaceae	0.60	Y	Biennial Herb
*Doronicum altaicum*	LC	Asteraceae	0.60	Y	Perennial Herb
*Arctium tomentosum*	LC	Asteraceae	0.60	Y	Biennial Herb
*Conium maculatum*	LC	Apiaceae	0.61	Y	Biennial Herb
*Oxytropis ochroleuca*	LC	Fabaceae	0.61	Y	Perennial Herb
*Roemeria refracta*	LC	Papaveraceae	0.61	Y	Annual herb
*Thymus marschallianus*	LC	Lamiaceae	0.61	Y	Semishrubs
*Thymus proximus*	LC	Lamiaceae	0.61	Y	Perennial Semishrubs
*Rosa laxa*	LC	Rosaceae	0.62	Y	Shrubs
*Prunus armeniaca*	NT	Rosaceae	0.63	N	Deciduous Tree
*Betula tianschanica*	LC	Betulaceae	0.63	N	Perennial Tree
*Glycyrrhiza uralensis*	NT	Fabaceae	0.65	N	Perennial Herb
*Rhodiola quadrifida*	NT	Crassulaceae	0.65	N	Perennial Herb
*Pseudolysimachion alatavicum*	NT	Plantaginaceae	0.69	Y	Perennial Herb
*Aconitum nemorum*	NT	Ranunculaceae	0.69	Y	Perennial Herb

**Table 4 plants-14-00099-t004:** List of primer pairs of barcoding genes used in species discrimination.

Barcode	Primer	Sequence
ITS	ITS_LEU	GTCCACTGAACCTTATCATTTAG
	ITS4	TCCTCCGCTTATTGATATGC
*matK*	*matK*472F	CCCRTYCATCTGGAAATCTTGGTTC
	*matK*1248R	GCTRTRATAATGAGAAAGATTTCTGC
*rbcL*	*rbcL*a_For	ATGTCACCACAAACAGAGACTAAAGC
	*rbcL*a_Rev	GTAAAATCAAGTCCACCRCG

**Table 5 plants-14-00099-t005:** Species assessment basis and reference for the AHP methodology.

Criterion Layer (C)	Index Layer (P)
Endangered Category (D1, 0.3938)	Extinct (9)
	Critically Endangered (7)
	Endangered (5)
	Vulnerable or Near Threatened (3)
	Least Concern (1)
Artificial planting (D2, 0.0438)	Unable to artificially breed (5)
	Artificial populations cannot form under natural conditions (3)
	Artificial populations can form under natural conditions (1)
Conservation Status (D3, 0.0127)	Not Conserved (7)
	Not Effectively Regulated (5)
	Conserved in situ, Translocated, or Isolated (3)
	Restoration of Populations (1)
Threat Persistence (D4, 0.0410)	Persistence of Threatening Factors (5)
	Transient Threatening Factors (3)
	Non-threatening Factors (1)
Domestic and international population impacts (D5, 0.0029)	Breeding from Abroad (5)
	No Effect within Domestic and Foreign populations (3)
	No Population Decline at home or abroad (1)
Species Trade Impacts (D6, 0.0059)	Over-utilization (3)
	No Over-utilization (1)
Community Status (D7, 0.0041)	Dominant Species (7)
	Subdominant Species (5)
	Companion Species (3)
	No Effect on Other Species in the Community (1)
Organism Type (D8, 0.0289)	Tree (7)
	Shrub (5)
	Perennial Herb (3)
	Annual Herb (1)
Endemic Species (D9, 0.0545)	Provincial Endemic Species (7)
	Regional Endemic Species (5)
	Chinese Endemic Species (3)
	Non-Chinese endemic Species (1)
Species Status (D10, 0.0060)	Monotypic Family (9)
	Oligocene Species (7)
	Monotypic Genus (5)
	Oligotypic Genus (3)
	Polyphyletic Genus (1)
The Status of Relict Species (D11, 0.0140)	Glacial Relict Plants (3)
	Non-Glacial Relict Plants (1)
Economic Values (D12, 0.0655)	Valuable for Development and Utilization (3)
	No Development or Utilization Value (1)
Hereditary Value (D13, 0.3270)	Unique utilization Value (5)
	Certain Utilization Value (3)
	No Utilization Value (1)

## Data Availability

Data are contained within the article and Appendix A.

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
