# Peer review of "Insights from DNA Barcodes-Based Phylogenetic Analysis of Medicinal Plants and Estimation of Their Conservation Status: A Case Study in the Tianshan Wild Forest, China"

_plants, 2025, doi:10.3390/plants14010099_

Round 1
Reviewer 1 Report
Comments and Suggestions for Authors
The manuscript submitted for review concerns the issue of insights from DNA barcodes-based phylogenetic analysis of medicinal plants and estimation of their conservation status. The research was conducted in the Tianshan Wild Forest, China. The authors emphasize that due to the centuries-old experience of Chinese natural medicine, this is a very important topic. The methodologies and insights presented in this study offer a valuable reference for DNA barcoding research on local specialty medicinal plants, as well as an evaluation framework for the protection of diversity and monitoring of genetic resources among medicinal plants in the area. It should be noted that the barcode method has been known for a long time. It is also recommended for planning activities related to the protection of valuable and endangered plant species.
In the introduction, the authors, in my opinion, focus too much on natural Chinese medicine. The fact of recognizing and protecting medicinal plants is, however, more important locally than universally. I suggest focusing more on the technique used. It would be good to change the passages about medicine and barcoding in places in the manuscript. First barcoding, second medicine.
Fig. 1 requires significant improvement. The presented map is too fragmentary. A more general map of China with the sampling location should be added, and only then this detailed fragment. Additionally, the subsequent elements in Fig. 1 are completely illegible, especially element "C". In general, I have to admit that all figures are hard to read (they are too small, there are too many of them). The authors should think about breaking these figures into more numerous ones. I rate fig. 6 positively, it's nice that the Authors decided to present the morphological characteristics of species ranked in Class 1 by hierarchical analysis 290 (AHP). Still, however, the photos are too small and therefore poorly legible. Moreover, the results are presented correctly and contribute new knowledge in the field of using the barcoding method.
The methods were appropriately selected, described and applied.
The conclusions are valid, although there is no clear conclusion regarding the conservation status of the plants.
Author Response
Comment 1: The manuscript submitted for review concerns the issue of insights from DNA barcodes-based phylogenetic analysis of medicinal plants and estimation of their conservation status. The research was conducted in the Tianshan Wild Forest, China. The authors emphasize that due to the centuries-old experience of Chinese natural medicine, this is a very important topic. The methodologies and insights presented in this study offer a valuable reference for DNA barcoding research on local specialty medicinal plants, as well as an evaluation framework for the protection of diversity and monitoring of genetic resources among medicinal plants in the area. It should be noted that the barcode method has been known for a long time. It is also recommended for planning activities related to the protection of valuable and endangered plant species.
Answer: For the valuable and endangered plant species according to the AHP method scores, we offered some advices based on the classed, which has been added to the results of 2.7 parts.
Comment 2: In the introduction, the authors, in my opinion, focus too much on natural Chinese medicine. The fact of recognizing and protecting medicinal plants is, however, more important locally than universally. I suggest focusing more on the technique used. It would be good to change the passages about medicine and barcoding in places in the manuscript. First barcoding, second medicine.
Answer: Based on your suggestions, we revised introductions: first barcoding, second medicine.
Comment 3: Fig. 1 requires significant improvement. The presented map is too fragmentary. A more general map of China with the sampling location should be added, and only then this detailed fragment. Additionally, the subsequent elements in Fig. 1 are completely illegible, especially element "C". In general, I have to admit that all figures are hard to read (they are too small, there are too many of them). The authors should think about breaking these figures into more numerous ones. I rate fig. 6 positively, it's nice that the Authors decided to present the morphological characteristics of species ranked in Class 1 by hierarchical analysis 290 (AHP). Still, however, the photos are too small and therefore poorly legible. Moreover, the results are presented correctly and contribute new knowledge in the field of using the barcoding method.
Answer: Based on your suggestions, we have made adjustments to all the pictures in the article, especially enlarging the font size to make it clearer.
The methods were appropriately selected, described and applied.
Comment 4: The conclusions are valid, although there is no clear conclusion regarding the conservation status of the plants.
Answer: In response to your opinions, we have graded the protection measures for these plants and drawn corresponding conclusions. The relevant additions are in Section 2.7 of the Results.

Reviewer 2 Report
Comments and Suggestions for Authors
The authors utilized DNA barcodes-based phylogenetic method for the identifying medicinal plants and estimated their conservation status. Three common DNA barcodes were used and some analytical methods for determining their current status. Their results were consistent with previous studies on the application of DNA barcodes, but how their conservation status was determined was not described clearly. In particular, introduction should be improved as it fails to provide enough background information, for example, AGBD and ASAP, should be explained for what they are, how different these two methods are. Also, these DNA barcodes are variable in their sequences above generic levels, phylogenetic analysis of all species together may result in inaccurate inference. I suggest analyzing the data with several subgroups based on their phylogenetic relatedness.
Here are my minor comments.
Lines 96 - 97: Proper literature should be cited.
Line 110: What is "an effective multilevel evaluation system"?
Lines 116 - 117: If plants belong to different families, it is unlikely to be misidentified. Need an explanation why the authors think like this.
Line 119: 89 herbaceous species, but 80 perennial herbs and 9 annual herbs?? Number does not match. Need to check.
Line 122: Need to define "Red List".
Lines 129 - 131: Why were habitat data investigated?
Lines 89 - 90: Need to give an example for "different methods may yield different results"
Lines 446 - 447: Need to explain where these information (P layer and C layer) were obtained.
Comments on the Quality of English Language
There were many grammatical errors, so this manuscript should be revised by someone who is fluent in English and scientific writing.
Author Response
Comment 1: The authors utilized DNA barcodes-based phylogenetic method for the identifying medicinal plants and estimated their conservation status. Three common DNA barcodes were used and some analytical methods for determining their current status. Their results were consistent with previous studies on the application of DNA barcodes, but how their conservation status was determined was not described clearly. In particular, introduction should be improved as it fails to provide enough background information, for example, AGBD and ASAP, should be explained for what they are, how different these two methods are.
Answer: we accepted the advice and based on your opinions, we have added analysis of similarities and differences between ASAP and ABGD and related content in the introduction.
Comment 2: Also, these DNA barcodes are variable in their sequences above generic levels, phylogenetic analysis of all species together may result in inaccurate inference. I suggest analyzing the data with several subgroups based on their phylogenetic relatedness.
Answer: While statistical and comparative analyses are crucial in DNA barcoding research, the species examined in this study encompass a diverse array of taxa from various families, leading to variability in the discriminatory power of subgroups for phylogenetic tree identification based on location. The performance of these analyses can differ significantly. In response to your inquiries, we propose employing a combination of multiple molecular identification methods and selecting analytical techniques flexibly to enhance research efficiency and accuracy. By judiciously utilizing statistical tools, we can evaluate the universal discriminability of a broad spectrum of species within the region, thereby providing a methodological framework for subsequent large-scale molecular identification efforts across diverse species. This approach will facilitate the development of an effective species identification and biodiversity assessment system.
Comment 3: Lines 96 - 97: Proper literature should be cited.
Answer: Cited
Comment 4: Line 110: What is "an effective multilevel evaluation system"?
Answer: It means “assessment should be thorough, involving multiple approaches or perspectives (such as ecological, economic, and social factors) to properly understand and address the situation”.
Comment 5: Lines 116 - 117: If plants belong to different families, it is unlikely to be misidentified. Need an explanation why the authors think like this.
Answer: Sentences were removed.
Comment 6:Line 119: 89 herbaceous species, but 80 perennial herbs and 9 annual herbs?? Number does not match. Need to check.
Answer: It was miscalculated. Corrected.
Comment 7:Line 122: Need to define "Red List".
Answer: We accepted your suggestion on the interpretation of the Red List and elaborated on the interpretation of the Red List and the latest release content in Section 2.1 of the results.
Comment 8: Lines 129 - 131: Why were habitat data investigated?
Answer: Actually, it is an additional information of samples collection sites. By this, we aimed to notice from which habitat types of plants were collected.
Comment 9: Lines 89 - 90: Need to give an example for "different methods may yield different results"
Answer: Thank you for your thoughtful advice. We found that the expression in this sentence was ambiguous and did not connect with the subsequent content, so we deleted this expression and elaborated on the similarities, differences, and advantages of different methods (ASAP and ABGD).
Comment 10: Lines 446 - 447: Need to explain where these information (P layer and C layer) were obtained.
Answer: Thank you for your careful reading. In fact, P layer is the abbreviation of Index layer, and C layer is the abbreviation of criterion layer. However, it is really inappropriate to put it this way and it may make readers think that there are other layers. Therefore, we have deleted the relevant words in the material and method section 4.4.
Comment 11: There were many grammatical errors, so this manuscript should be revised by someone who is fluent in English and scientific writing.
Answer: we have carefully revised the sentences gramma. And foreigner export Tohir A. Bozorov checked all the writing.

Round 2
Reviewer 2 Report
Comments and Suggestions for Authors
The authors employed three DNA barcodes to identify medicinal plants in a specific region and evaluated their status using various methods. While the revised manuscript represents a significant improvement over the initial version, several points require clarification before a final decision can be made.
For instance, the authors do not explain why multiloci markers performed worse than a single marker, despite the general expectation that multiloci markers exhibit superior performance. Additionally, the authors should clarify how they addressed gaps in the alignments of each marker. For example, ITS regions are highly variable in length, often leading to ambiguous alignments across different groups. Furthermore, ITS regions frequently exhibit heterogeneous sequences, necessitating cloning to obtain clean sequences. However, the manuscript does not discuss this issue. Given that the authors sequenced 101 species, they likely encountered this challenge and should detail their approach to resolving it.
Please also consider the additional comments provided in the attached file.

Author Response
The reviewer’s comment:
The authors employed three DNA barcodes to identify medicinal plants in a specific region and evaluated their status using various methods. While the revised manuscript represents a significant improvement over the initial version, several points require clarification before a final decision can be made.
Thank you for your good comments and suggestions and for reading the last version.
Comment 1: For instance, the authors do not explain why multiloci markers performed worse than a single marker, despite the general expectation that multiloci markers exhibit superior performance.
Answer: Thank you for taking an interest in our article and your thoughtful comments. We accept your advice and have added an explaination in the discusion part line 497, the content as follow “This difference may be due to the wide range of species involved in this study. When identifying species, the lack of species reference sequences in relevant public databases resulted in incomplete data or insufficient sample coverage for the ITS+matK+rbcL combination. Combinations may not cover variation in all species, resulting in reduced discriminatory power.”
Comment 2: Additionally, the authors should clarify how they addressed gaps in the alignments of each marker. For example, ITS regions are highly variable in length, often leading to ambiguous alignments across different groups. Furthermore, ITS regions frequently exhibit heterogeneous sequences, necessitating cloning to obtain clean sequences. However, the manuscript does not discuss this issue. Given that the authors sequenced 101 species, they likely encountered this challenge and should detail their approach to resolving it.
Answer: Thank you for your thoughtful suggestions. In fact, because this study involves many species, families, and genera, we do encounter the problems you mentioned when dealing with such highly variable areas in different locations. For comparison issues, we use Geneious to provide a variety of comparison algorithms. We choosed the appropriate algorithm based on the size and characteristics of the data set. On this basis, manual sequence trimming was performed on regions with very small length variations that made it difficult to align, thereby improving the accuracy of alignment. And we added the relative content in the materials and methods 4.3 line 598, as followed “Given the wide range of species included in the analysis, the MAFFT algorithm in Geneious Prime, which is suitable for processing large data sets, was used to facilitate sequence alignment using various algorithms. In addition, manual sequence pruning was performed on regions that presented alignment challenges, thereby improving the accuracy of the comparison.”
Comment 3: Line 51 and line 53 Need to explain whether these are the same or different areas each other.
Answer: trnH-psbA and psbA-trnH refer to the same region, so we revised the line 53 (now line 64) as trnH-psbA.
Comment 4: Line57 and Line 58 ITS non-italic for ITS
Answer: we revised 'ITS' to non-italic format(now in line 69 and 70).
Comment 5: line 67 what is this?
Answer: we have added the full name in line 79 now “molecular operational taxonomic units (MOTUs)”.
Comment 6: line 69 what kind of distribution? Geographical or phylogenetical?
Answer: Thanks. It refers to phylogenetic distribution, and we have inserted “phylogenetic” before “distribution” in line 82 now.
Comment 7: line 100 Need to explain what it is and how it is different from two methods mentioned above.
Answer: We have given a detailed explanation of the AHP method. The ASAP and ABGD methods are used for species identification, and the AHP method is used for species protection and utilization status assessment methods. Therefore, based on your feedback, we have made the following modifications: “In terms of species protection decision-making, a comprehensive evaluation system for plant protection and utilization is built based on the AHP analytic hierarchy process to quantify and compare different factors of species from multiple dimensions such as biological characteristics, ornamental characteristics, and development potential [32]. Based on the quantified comprehensive evaluation of plants The evaluation values ​​are graded to make accurate and effective development and utilization decisions for plants of different grades.” in line 113 now.
Comment 8: in line 121 and 130, the higher plant what does it mean?
Answer: We have modified it to “terrestrial plants” in both lines 140 and 149 now.
Comment 9: line 133 “4” and “2” changed as “four ” and “two”.
Answer: We rechecked the number of plants and found that it should be changed to ‘five’ and ‘two’ in line 153.
Comment 10: in line 146 what about DD?
Answer: We have added the DD meaning of ‘DD’ as ‘data deficient’ in line 166 now.
Comment 11: in line 153 Table 2 instead of table 3.
Answer: Table 3 was revised as Table 2 in line 173.
Comment 12: line 173 what kind of distances? Need to specify.
Answer: The model (optimal model was calculated based on 'best model criterion' in MEGA11. We found that this sentence was ambiguous here, so we deleted it in line 193.
Comment 13:In the line 186 which one?
Answer:It was ITS locus, and we replaced the certain loci as ITS locus in line 206 now.
Comment 14: In the line 324 How were these three classes defined? What are the meanings of these classes?
Answer: The classes were defined in Table S2. At the end of this sentence, the source is inserted as (Table S2,Table S7) in line 344.
Comment 15: In the line 338 need more explanation for this.
Answer: Based on your opinions, we have added the relevant explanation as “Medicinal plants, including Rheum wittrockii, Aquilegia atrovinosa, Cynoglossum officinale, Glycyrrhiza uralensis, and Rhodiola quadrifida, possess distinct medicinal properties and active compounds, contributing significantly to the economic value of pharmaceutical and healthcare products.” in line 358.
Comment 16: In line 342 need more explanation for this.
Answer: We added the explanation as “including population genetic structure, genetic diversity of germplasm resources, genetic differentiation and adaptability, etc.” in line 365 now.
Comment 17: In the line 343 Should be moved to discussion.
Answer: The content was placed in the discussion line 522 now.
Comment 18: In the line 425 and 426 the “9” and “3” changed as “nine ” and “three”.
Answer: They have been changed as “nine ” and “three” in line 449 and 450 now.
Comment 19: In the line 473 how does species specificity affect the performance of multiloci markers? Need more explanation.
Answer: The explanation of this part is as follows “This difference may be due to the wide range of species involved in this study. When identifying species, the lack of species reference sequences in relevant public databases resulted in incomplete data or insufficient sample coverage for the ITS+matK+rbcL combination. Combinations may not cover variation in all species, resulting in reduced discriminatory power.” in line 497 now.
Comment 20: In the line 490 and 491, should changed as “AHP”.
Answer: We haved replaced “Analytical Hierarchy Process (AHP)” as “AHP” in line 508 now.
Comment 21: In the line 541 Changed as “ABI 3730xl”.
Answer: The phrase was changed as “ABI 3730xl” in line 589 now.
Additional Changes:
We have also made additional changes to the manuscript that were not specifically pointed out by the reviewers but which we believe improve the quality and clarity of the paper. These changes include:
- We have checked the full text of the statistics on the number of various types of plants and corrected any errors.
- In the introduction section, we have added some literature content so that readers can better understand the related research on DNA barcoding.
